# Are Sixteen Heads Really Better than One?

**Paul Michel**
Language Technologies Institute
Carnegie Mellon University
Pittsburgh, PA
`pmichel1@cs.cmu.edu`

**Omer Levy**
Facebook Artificial Intelligence Research
Seattle, WA
`omerlevy@fb.com`

**Graham Neubig**
Language Technologies Institute
Carnegie Mellon University
Pittsburgh, PA
`gneubig@cs.cmu.edu`

## Abstract

Attention is a powerful and ubiquitous mechanism for allowing neural models to focus on particular salient pieces of information by taking their weighted average when making predictions. In particular, *multi-headed attention* is a driving force behind many recent state-of-the-art natural language processing (NLP) models such as Transformer-based MT models and BERT. These models apply multiple attention mechanisms in parallel, with each attention "head" potentially focusing on different parts of the input, which makes it possible to express sophisticated functions beyond the simple weighted average. In this paper we make the surprising observation that even if models have been trained using multiple heads, in practice, a large percentage of attention heads can be removed at test time without significantly impacting performance. In fact, some layers can even be reduced to a single head. We further examine greedy algorithms for pruning down models, and the potential speed, memory efficiency, and accuracy improvements obtainable therefrom. Finally, we analyze the results with respect to which parts of the model are more reliant on having multiple heads, and provide precursory evidence that training dynamics play a role in the gains provided by multi-head attention[1].

## 1   Introduction

Transformers (Vaswani et al., 2017) have shown state of the art performance across a variety of NLP tasks, including, but not limited to, machine translation (Vaswani et al., 2017; Ott et al., 2018), question answering (Devlin et al., 2018), text classification (Radford et al., 2018), and semantic role labeling (Strubell et al., 2018). Central to its architectural improvements, the Transformer extends the standard attention mechanism (Bahdanau et al., 2015; Cho et al., 2014) via multi-headed attention (MHA), where attention is computed independently by $N_h$ parallel attention mechanisms (heads). It has been shown that beyond improving performance, MHA can help with subject-verb agreement (Tang et al., 2018) and that some heads are predictive of dependency structures (Raganato and Tiedemann, 2018). Since then, several extensions to the general methodology have been proposed (Ahmed et al., 2017; Shen et al., 2018).

However, it is still not entirely clear: what do the multiple heads in these models buy us? In this paper, we make the surprising observation that – in both Transformer-based models for machine translation and BERT-based (Devlin et al., 2018) natural language inference – most attention heads can be individually removed after training without any significant downside in terms of test performance (§3.2). Remarkably, many attention layers can even be individually reduced to a single attention head without impacting test performance (§3.3).

Based on this observation, we further propose a simple algorithm that greedily and iteratively prunes away attention heads that seem to be contributing less to the model. By jointly removing attention heads from the entire network, without restricting pruning to a single layer, we find that large parts of the network can be removed with little to no consequences, but that the majority of heads must remain to avoid catastrophic drops in performance (§4). We further find that this has significant benefits for inference-time efficiency, resulting in up to a 17.5% increase in inference speed for a BERT-based model.

We then delve into further analysis. A closer look at the case of machine translation reveals that the encoder-decoder attention layers are particularly sensitive to pruning, much more than the self-attention layers, suggesting that multi-headedness plays a critical role in this component (§5). Finally, we provide evidence that the distinction between important and unimporant heads increases as training progresses, suggesting an interaction between multi-headedness and training dynamics (§6).

## 2 Background: Attention, Multi-headed Attention, and Masking

In this section we lay out the notational groundwork regarding attention, and also describe our method for masking out attention heads.

### 2.1 Single-headed Attention

We briefly recall how vanilla attention operates. We focus on scaled bilinear attention (Luong et al., 2015), the variant most commonly used in MHA layers. Given a sequence of $n$ $d$-dimensional vectors $\mathbf{x} = x_1, \ldots, x_n \in \mathbb{R}^d$, and a query vector $q \in \mathbb{R}^d$, the attention layer parametrized by $W_k, W_q, W_v, W_o \in \mathbb{R}^{d \times d}$ computes the weighted sum:

$$\text{Att}_{W_k, W_q, W_v, W_o}(\mathbf{x}, q) = W_o \sum_{i=1}^{n} \alpha_i W_v x_i$$

$$\text{where } \alpha_i = \text{softmax} \left( \frac{q^\intercal W_q^\intercal W_k x_i}{\sqrt{d}} \right)$$

In self-attention, every $x_i$ is used as the query $q$ to compute a new sequence of representations, whereas in sequence-to-sequence models $q$ is typically a decoder state while $\mathbf{x}$ corresponds to the encoder output.

### 2.2 Multi-headed Attention

In *multi-headed* attention (MHA), $N_h$ independently parameterized attention layers are applied in parallel to obtain the final result:

$$\text{MHAtt}(\mathbf{x}, q) = \sum_{h=1}^{N_h} \text{Att}_{W_k^h, W_q^h, W_v^h, W_o^h}(\mathbf{x}, q) \tag{1}$$

where $W_k^h, W_q^h, W_v^h \in \mathbb{R}^{d_h \times d}$ and $W_o^h \in \mathbb{R}^{d \times d_h}$. When $d_h = d$, MHA is strictly more expressive than vanilla attention. However, to keep the number of parameters constant, $d_h$ is typically set to $\frac{d}{N_h}$, in which case MHA can be seen as an ensemble of low-rank vanilla attention layers[2]. In the following, we use $\text{Att}_h(x)$ as a shorthand for the output of head $h$ on input $x$.

To allow the different attention heads to interact with each other, transformers apply a non-linear feed-forward network over the MHA's output, at each transformer layer (Vaswani et al., 2017).

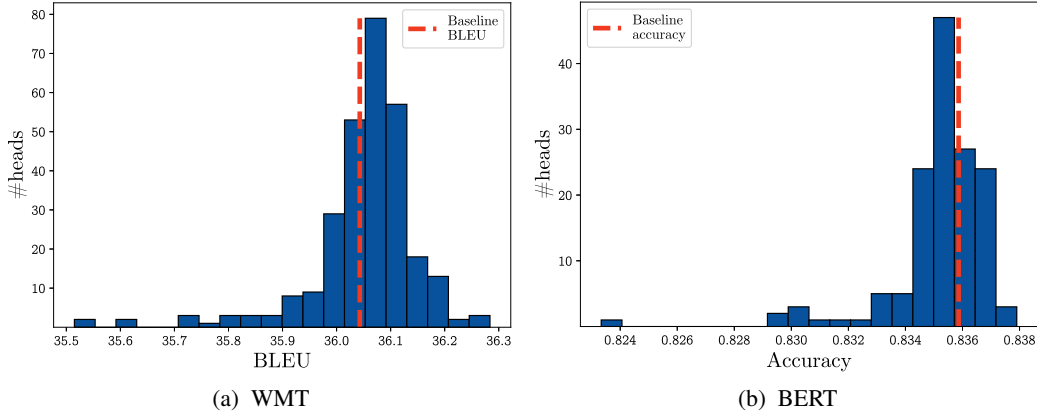

|            | (a) WMT | (b) BERT |
|------------|---------|----------|

Figure 1: Distribution of heads by model score after masking.

## 2.3 Masking Attention Heads

In order to perform ablation experiments on the heads, we modify the formula for MHAtt:

$$\text{MHAtt}(\mathbf{x}, q) = \sum_{h=1}^{N_h} \xi_h \text{Att}_{W_k^h, W_q^h, W_v^h, W_o^h}(\mathbf{x}, q)$$

where the $\xi_h$ are mask variables with values in $\{0, 1\}$. When all $\xi_h$ are equal to 1, this is equivalent to the formulation in Equation 1. In order to mask head $h$, we simply set $\xi_h = 0$.

## 3  Are All Attention Heads Important?

We perform a series of experiments in which we remove one or more attention heads from a given architecture at test time, and measure the performance difference. We first remove a single attention head at each time (§3.2) and then remove every head in an entire layer except for one (§3.3).

## 3.1  Experimental Setup

In all following experiments, we consider two trained models:

**WMT**   This is the original "large" transformer architecture from Vaswani et al. 2017 with 6 layers and 16 heads per layer, trained on the WMT2014 English to French corpus. We use the pretrained model of Ott et al. 2018.[3] and report BLEU scores on the `newstest2013` test set. In accordance with Ott et al. 2018, we compute BLEU scores on the tokenized output of the model using Moses (Koehn et al., 2007). Statistical significance is tested with paired bootstrap resampling (Koehn, 2004) using `compare-mt`[4] (Neubig et al., 2019) with 1000 resamples. A particularity of this model is that it features 3 distinct attention mechanism: encoder self-attention (Enc-Enc), encoder-decoder attention (Enc-Dec) and decoder self-attention (Dec-Dec), all of which use MHA.

**BERT**   BERT (Devlin et al., 2018) is a single transformer pre-trained on an unsupervised cloze-style "masked language modeling task" and then fine-tuned on specific tasks. At the time of its inception, it achieved state-of-the-art performance on a variety of NLP tasks. We use the pre-trained `base-uncased` model of Devlin et al. 2018 with 12 layers and 12 attention heads which we fine-tune and evaluate on MultiNLI (Williams et al., 2018). We report accuracies on the "matched" validation set. We test for statistical significance using the t-test. In contrast with the WMT model, BERT only features one attention mechanism (self-attention in each layer).

## 3.2  Ablating One Head

To understand the contribution of a particular attention head $h$, we evaluate the model's performance while masking that head (i.e. replacing $Att_h(x)$ with zeros). If the performance sans $h$ is significantly

| Head<br>Layer | 1 | 2 | 3 | 4 | 5 | 6 | 7 | 8 | 9 | 10 | 11 | 12 | 13 | 14 | 15 | 16 |
|---|---|---|---|---|---|---|---|---|---|---|---|---|---|---|---|---|
| 1 | 0.03 | 0.07 | 0.05 | -0.06 | 0.03 | **-0.53** | 0.09 | **-0.33** | 0.06 | 0.03 | 0.11 | 0.04 | 0.01 | -0.04 | 0.04 | 0.00 |
| 2 | 0.01 | 0.04 | 0.10 | **0.20** | 0.06 | 0.03 | 0.00 | 0.09 | 0.10 | 0.04 | **0.15** | 0.03 | 0.05 | 0.04 | 0.14 | 0.04 |
| 3 | 0.05 | -0.01 | 0.08 | 0.09 | 0.11 | 0.02 | 0.03 | 0.03 | -0.00 | 0.13 | 0.09 | 0.09 | -0.11 | **0.24** | 0.07 | -0.04 |
| 4 | -0.02 | 0.03 | 0.13 | 0.06 | -0.05 | 0.13 | 0.14 | 0.05 | 0.02 | 0.14 | 0.05 | 0.06 | 0.03 | -0.06 | -0.10 | -0.06 |
| 5 | **-0.31** | -0.11 | -0.04 | 0.12 | 0.10 | 0.02 | 0.09 | 0.08 | 0.04 | **0.21** | -0.02 | 0.02 | -0.03 | -0.04 | 0.07 | -0.02 |
| 6 | 0.06 | 0.07 | **-0.31** | 0.15 | -0.19 | 0.15 | 0.11 | 0.05 | 0.01 | -0.08 | 0.06 | 0.01 | 0.01 | 0.02 | 0.07 | 0.05 |

Table 1: Difference in BLEU score for each head of the encoder's self attention mechanism. Underlined numbers indicate that the change is statistically significant with $p < 0.01$. The base BLEU score is 36.05.

| Layer | Enc-Enc | Enc-Dec | Dec-Dec |
|---|---|---|---|
| 1 | **-1.31** | **0.24** | -0.03 |
| 2 | -0.16 | 0.06 | 0.12 |
| 3 | 0.12 | 0.05 | 0.18 |
| 4 | -0.15 | -0.24 | 0.17 |
| 5 | 0.02 | **-1.55** | -0.04 |
| 6 | **-0.36** | **-13.56** | 0.24 |

Table 2: Best delta BLEU by layer when only one head is kept in the WMT model. Underlined numbers indicate that the change is statistically significant with $p < 0.01$.

| Layer | | Layer | |
|---|---|---|---|
| 1 | -0.01% | 7 | 0.05% |
| 2 | 0.10% | 8 | -0.72% |
| 3 | -0.14% | 9 | -0.96% |
| 4 | -0.53% | 10 | 0.07% |
| 5 | -0.29% | 11 | -0.19% |
| 6 | -0.52% | 12 | -0.12% |

Table 3: Best delta accuracy by layer when only one head is kept in the BERT model. None of these results are statistically significant with $p < 0.01$.

worse than the full model's, $h$ is obviously important; if the performance is comparable, $h$ is redundant given the rest of the model.

Figures 1a and 1b shows the distribution of heads in term of the model's score after masking it, for WMT and BERT respectively. We observe that the majority of attention heads can be removed without deviating too much from the original score. Surprisingly, in some cases removing an attention head results in an increase in BLEU/accuracy.

To get a finer-grained view on these results we zoom in on the encoder self-attention layers of the WMT model in Table 1. Notably, we see that only 8 (out of 96) heads cause a statistically significant change in performance when they are removed from the model, half of which actually result in a higher BLEU score. This leads us to our first observation: **at test time, most heads are redundant given the rest of the model**.

### 3.3 Ablating All Heads but One

This observation begets the question: is more than one head even needed? Therefore, we compute the difference in performance when all heads except one are removed, within a single layer. In Table 2 and Table 3 we report the best score for each layer in the model, i.e. the score when reducing the entire layer to the single most important head.

We find that, for most layers, one head is indeed sufficient at test time, even though the network was trained with 12 or 16 attention heads. This is remarkable because these layers can be reduced to single-headed attention with only $1/16$th (resp. $1/12$th) of the number of parameters of a vanilla attention layer. However, some layers *do* require multiple attention heads; for example, substituting the last layer in the encoder-decoder attention of WMT with a single head degrades performance by at least 13.5 BLEU points. We further analyze when different modeling components depend on more heads in §5.

Additionally, we verify that this result holds even when we don't have access to the evaluation set when selecting the head that is "best on its own". For this purpose, we select the best head for each layer on a validation set (`newstest2013` for WMT and a 5,000-sized randomly selected subset of the training set of MNLI for BERT) and evaluate the model's performance on a test set

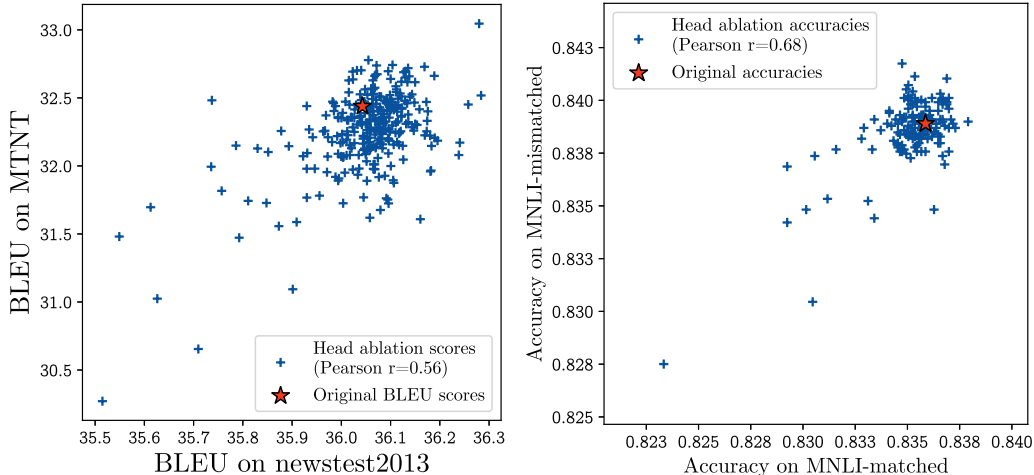

(a) BLEU on `newstest2013` and MTNT when individual heads are removed from WMT. Note that the ranges are not the same one the X and Y axis as there seems to be much more variation on MTNT.

(b) Accuracies on MNLI-matched and -mismatched when individual heads are removed from BERT. Here the scores remain in the same approximate range of values.

Figure 2: Cross-task analysis of effect of pruning on accuracy

(`newstest2014` for WMT and the MNLI-matched validation set for BERT). We observe that similar findings hold: keeping only one head does not result in a statistically significant change in performance for 50% (resp. 100%) of layers of WMT (resp. BERT). The detailed results can be found in Appendix A.

## 3.4   Are Important Heads the Same Across Datasets?

There is a caveat to our two previous experiments: these results are only valid on specific (and rather small) test sets, casting doubt on their generalizability to other datasets. As a first step to understand whether some heads are universally important, we perform the same ablation study on a second, out-of-domain test set. Specifically, we consider the MNLI "mismatched" validation set for BERT and the MTNT English to French test set (Michel and Neubig, 2018) for the WMT model, both of which have been assembled for the very purpose of providing contrastive, out-of-domain test suites for their respective tasks.

We perform the same ablation study as §3.2 on each of these datasets and report results in Figures 2a and 2b. We notice that there is a positive, $> 0.5$ correlation ($p < 001$) between the effect of removing a head on both datasets. Moreover, heads that have the highest effect on performance on one domain tend to have the same effect on the other, which suggests that the most important heads from §3.2 are indeed "universally" important.

## 4   Iterative Pruning of Attention Heads

In our ablation experiments (§3.2 and §3.3), we observed the effect of removing one or more heads within a single layer, without considering what would happen if we altered two or more different layers at the same time. To test the compounding effect of pruning multiple heads from across the entire model, we sort all the attention heads in the model according to a proxy importance score (described below), and then remove the heads one by one. We use this iterative, heuristic approach to avoid combinatorial search, which is impractical given the number of heads and the time it takes to evaluate each model.

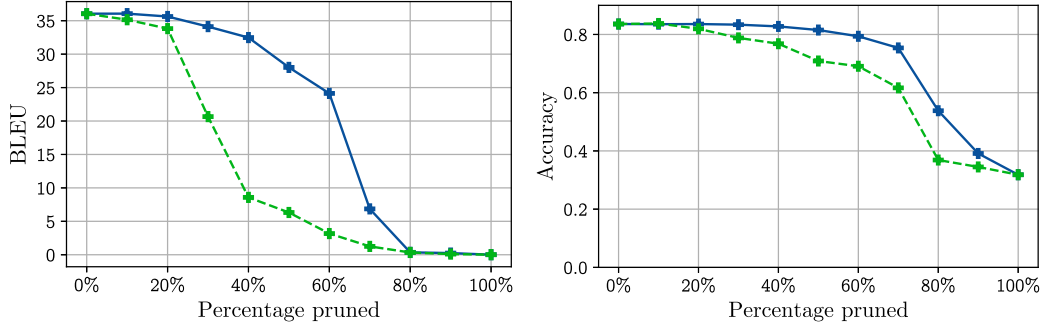

(a) Evolution of BLEU score on `newstest2013` when heads are pruned from WMT.

(b) Evolution of accuracy on the MultiNLI-matched validation set when heads are pruned from BERT.

Figure 3: Evolution of accuracy by number of heads pruned according to $I_h$ (solid blue) and individual oracle performance difference (dashed green).

## 4.1 Head Importance Score for Pruning

As a proxy score for head importance, we look at the expected sensitivity of the model to the mask variables $\xi_h$ defined in §2.3:

$$I_h = \mathbb{E}_{x \sim X} \left| \frac{\partial \mathcal{L}(x)}{\partial \xi_h} \right| \tag{2}$$

where $X$ is the data distribution and $\mathcal{L}(x)$ the loss on sample $x$. Intuitively, if $I_h$ has a high value then changing $\xi_h$ is liable to have a large effect on the model. In particular we find the absolute value to be crucial to avoid datapoints with highly negative or positive contributions from nullifying each other in the sum. Plugging Equation 1 into Equation 2 and applying the chain rule yields the following final expression for $I_h$:

$$I_h = \mathbb{E}_{x \sim X} \left| \text{Att}_h(x)^T \frac{\partial \mathcal{L}(x)}{\partial \text{Att}_h(x)} \right|$$

This formulation is reminiscent of the wealth of literature on pruning neural networks (LeCun et al., 1990; Hassibi and Stork, 1993; Molchanov et al., 2017, inter alia). In particular, it is equivalent to the Taylor expansion method from Molchanov et al. (2017).

As far as performance is concerned, estimating $I_h$ only requires performing a forward and backward pass, and therefore is not slower than training. In practice, we compute the expectation over the training data or a subset thereof.[5] As recommended by Molchanov et al. (2017) we normalize the importance scores by layer (using the $\ell_2$ norm).

## 4.2 Effect of Pruning on BLEU/Accuracy

Figures 3a (for WMT) and 3b (for BERT) describe the effect of attention-head pruning on model performance while incrementally removing $10\%$ of the total number of heads in order of increasing $I_h$ at each step. We also report results when the pruning order is determined by the score difference from §3.2 (in dashed lines), but find that using $I_h$ is faster and yields better results.

We observe that this approach allows us to prune up to $20\%$ and $40\%$ of heads from WMT and BERT (respectively), without incurring any noticeable negative impact. Performance drops sharply when pruning further, meaning that neither model can be reduced to a purely single-head attention model without retraining or incurring substantial losses to performance. We refer to Appendix B for experiments on four additional datasets.

### 4.3 Effect of Pruning on Efficiency

Beyond the downstream task performance, there are intrinsic advantages to pruning heads. First, each head represents a non-negligible proportion of the total parameters in each attention layer ($6.25\%$ for WMT, $\approx 8.34\%$ for BERT), and thus of the total model (roughly speaking, in both our models, approximately one third of the total number of parameters is devoted to MHA across all layers).[6] This is appealing in the context of deploying models in memory-constrained settings.

| Batch size | 1 | 4 | 16 | 64 |
|---|---|---|---|---|
| Original | $17.0 \pm 0.3$ | $67.3 \pm 1.3$ | $114.0 \pm 3.6$ | $124.7 \pm 2.9$ |
| Pruned (50%) | $17.3 \pm 0.6$ | $69.1 \pm 1.3$ | $134.0 \pm 3.6$ | $146.6 \pm 3.4$ |
| | (+1.9%) | (+2.7%) | (+17.5%) | (+17.5%) |

Table 4: Average inference speed of BERT on the MNLI-matched validation set in examples per second ($\pm$ standard deviation). The speedup relative to the original model is indicated in parentheses.

Moreover, we find that actually pruning the heads (and not just masking) results in an appreciable increase in inference speed. Table 4 reports the number of examples per second processed by BERT, before and after pruning 50% of all attention heads. Experiments were conducted on two different machines, both equipped with GeForce GTX 1080Ti GPUs. Each experiment is repeated 3 times on each machine (for a total of 6 datapoints for each setting). We find that pruning half of the model's heads speeds up inference by up to $\approx 17.5\%$ for higher batch sizes (this difference vanishes for smaller batch sizes).

## 5 When Are More Heads Important? The Case of Machine Translation

As shown in Table 2, not all MHA layers can be reduced to a single attention head without significantly impacting performance. To get a better idea of how much each part of the transformer-based translation model relies on multi-headedness, we repeat the heuristic pruning experiment from §4 for each type of attention separately (Enc-Enc, Enc-Dec, and Dec-Dec).

Figure 4 shows that performance drops much more rapidly when heads are pruned from the Enc-Dec attention layers. In particular, pruning more than $60\%$ of the Enc-Dec attention heads will result in catastrophic performance degradation, while the encoder and decoder self-attention layers can still produce reasonable translations (with BLEU scores around 30) with only 20% of the original attention heads. In other words, encoder-decoder attention is much more dependent on multi-headedness than self-attention.

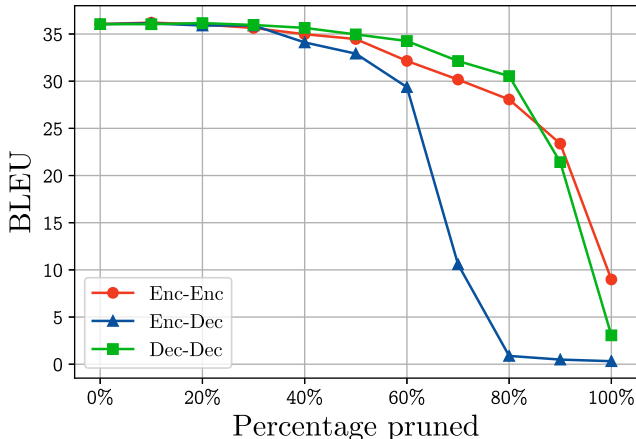

Figure 4: BLEU when incrementally pruning heads from each attention type in the WMT model.

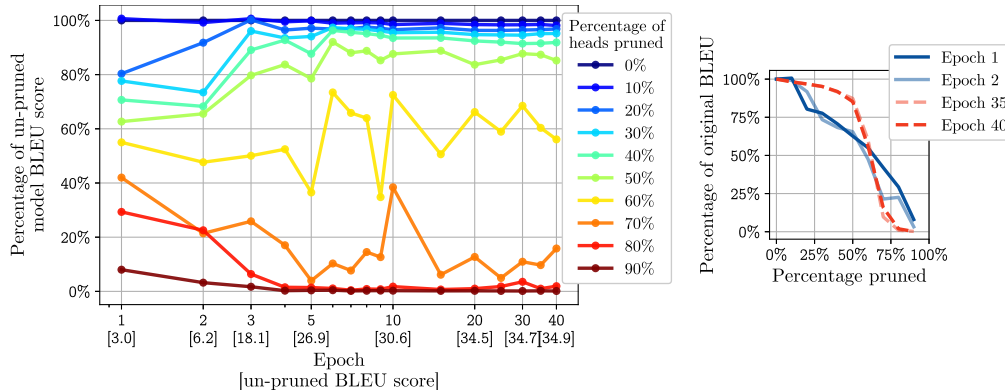

Figure 5: **Left side**: relationship between percentage of heads pruned and relative score decrease during training of the IWSLT model. We report epochs on a logarithmic scale. The BLEU score of the original, un-pruned model is indicated in brackets. **Right side**: focus on the difference in behaviour at the beginning (epochs 1 and 2) and end (epochs 35 and 40) of training.

## 6 Dynamics of Head Importance during Training

Previous sections tell us that some heads are more important than others in *trained* models. To get more insight into the dynamics of head importance *during training*, we perform the same incremental pruning experiment of §4.2 at every epoch. We perform this experiment on a smaller version of WMT model (6 layers and 8 heads per layer), trained for German to English translation on the smaller IWSLT 2014 dataset Cettolo et al. (2015). We refer to this model as **IWSLT**.

Figure 5 reports, for each level of pruning (by increments of 10% — 0% corresponding to the original model), the evolution of the model's score (on `newstest2013`) for each epoch. For better readability we display epochs on a logarithmic scale, and only report scores every 5 epochs after the 10th). To make scores comparable across epochs, the Y axis reports the relative degradation of BLEU score with respect to the un-pruned model at each epoch. Notably, we find that there are two distinct regimes: in very early epochs (especially 1 and 2), performance decreases linearly with the pruning percentage, *i.e.* the relative decrease in performance is independent from $I_h$, indicating that most heads are more or less equally important. From epoch 10 onwards, there is a concentration of unimportant heads that can be pruned while staying within $85-90\%$ of the original BLEU score (up to 40% of total heads).

This suggests that the important heads are determined early (but not immediately) during the training process. The two phases of training are reminiscent of the analysis by Shwartz-Ziv and Tishby (2017), according to which the training of neural networks decomposes into an "empirical risk minimization" phase, where the model maximizes the mutual information of its intermediate representations with the labels, and a "compression" phase where the mutual information with the input is minimized. A more principled investigation of this phenomenon is left to future work.

## 7 Related work

The use of an attention mechanism in NLP and in particular neural machine translation (NMT) can be traced back to Bahdanau et al. (2015) and Cho et al. (2014), and most contemporaneous implementations are based on the formulation from Luong et al. (2015). Attention was shortly adapted (successfully) to other NLP tasks, often achieving then state-of-the-art performance in reading comprehension (Cheng et al., 2016), natural language inference (Parikh et al., 2016) or abstractive summarization (Paulus et al., 2017) to cite a few. Multi-headed attention was first introduced by Vaswani et al. (2017) for NMT and English constituency parsing, and later adopted for transfer learning (Radford et al., 2018; Devlin et al., 2018), language modeling (Dai et al., 2019; Radford et al., 2019), or semantic role labeling (Strubell et al., 2018), among others.

There is a rich literature on pruning trained neural networks, going back to LeCun et al. (1990) and Hassibi and Stork (1993) in the early 90s and reinvigorated after the advent of deep learning, with two

orthogonal approaches: fine-grained "weight-by-weight" pruning (Han et al., 2015) and structured pruning (Anwar et al., 2017; Li et al., 2016; Molchanov et al., 2017), wherein entire parts of the model are pruned. In NLP , structured pruning for auto-sizing feed-forward language models was first investigated by Murray and Chiang (2015). More recently, fine-grained pruning approaches have been popularized by See et al. (2016) and Kim and Rush (2016) (mostly on NMT).

Concurrently to our own work, Voita et al. (2019) have made to a similar observation on multi-head attention. Their approach involves using LRP (Binder et al., 2016) for determining important heads and looking at specific properties such as attending to adjacent positions, rare words or syntactically related words. They propose an alternate pruning mechanism based on doing gradient descent on the mask variables $\xi_h$. While their approach and results are complementary to this paper, our study provides additional evidence of this phenomenon beyond NMT, as well as an analysis of the training dynamics of pruning attention heads.

# 8   Conclusion

We have observed that MHA does not always leverage its theoretically superior expressiveness over vanilla attention to the fullest extent. Specifically, we demonstrated that in a variety of settings, several heads can be removed from trained transformer models without statistically significant degradation in test performance, and that some layers can be reduced to only one head. Additionally, we have shown that in machine translation models, the encoder-decoder attention layers are much more reliant on multi-headedness than the self-attention layers, and provided evidence that the relative importance of each head is determined in the early stages of training. We hope that these observations will advance our understanding of MHA and inspire models that invest their parameters and attention more efficiently.

# Acknowledgments

The authors would like to extend their thanks to the anonymous reviewers for their insightful feedback. We are also particularly grateful to Thomas Wolf from Hugging Face, whose independent reproduction efforts allowed us to find and correct a bug in our speed comparison experiments. This research was supported in part by a gift from Facebook.

## Footnotes

[1]Code to replicate our experiments is provided at `https://github.com/pmichel31415/are-16-heads-really-better-than-1`

[2]This notation, equivalent to the "concatenation" formulation from Vaswani et al. (2017), is used to ease exposition in the following sections.

[3]`https://github.com/pytorch/fairseq/tree/master/examples/translation`

[4]`https://github.com/neulab/compare-mt`

[5]For the WMT model we use all `newstest20[09-12]` sets to estimate $I$.

[6]Slightly more in WMT because of the Enc-Dec attention.

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
