[Supplementary Material]

# A   Ablating All Heads but One: Additional Experiment.

Tables 5 and 6 report the difference in performance when only one head is kept for any given layer. The head is chosen to be the best head on its own on a *separate* dataset.

| Layer | Enc-Enc | Enc-Dec | Dec-Dec |
|---|---|---|---|
| 1 | **-1.96** | 0.02 | 0.03 |
| 2 | **-0.57** | 0.09 | -0.13 |
| 3 | **-0.45** | **-0.42** | 0.00 |
| 4 | -0.30 | **-0.60** | -0.31 |
| 5 | -0.32 | **-2.75** | **-0.66** |
| 6 | **-0.67** | **-18.89** | -0.03 |

| Layer | | | Layer | |
|---|---|---|---|---|
| 1 | -0.01% | 7 | 0.05% | |
| 2 | -0.02% | 8 | -0.72% | |
| 3 | -0.26% | 9 | -0.96% | |
| 4 | -0.53% | 10 | 0.07% | |
| 5 | -0.29% | 11 | -0.19% | |
| 6 | -0.52% | 12 | -0.15% | |

Table 5: Best delta BLEU by layer on `newstest2014` when only the best head (as evaluated on `newstest2013`) is kept in the WMT model. Underlined numbers indicate that the change is statistically significant with $p < 0.01$.

Table 6: Best delta accuracy by layer on the validation set of MNLI-matched when only the best head (as evaluated on 5,000 training examples) is kept in the BERT model. None of these results are statistically significant with $p < 0.01$.

# B   Additional Pruning Experiments

We report additional results for the importance-driven pruning approach from Section 4 on 4 additional datasets:

- **SST-2**: The GLUE version of the Stanford Sentiment Treebank (Socher et al., 2013). We use a fine-tuned BERT as our model.

- **CoLA**: The GLUE version of the Corpus of Linguistic Acceptability (Warstadt et al., 2018). We use a fine-tuned BERT as our model.

- **MRPC**: The GLUE version of the Microsoft Research Paraphrase Corpus (Dolan and Brockett, 2005). We use a fine-tuned BERT as our model.

- **IWSLT**: The German to English translation dataset from IWSLT 2014 (Cettolo et al., 2015). We use the same smaller model described in Section 6.

Figure 6 shows that in some cases up to 60% (SST-2) or 50% (CoLA, MRPC) of heads can be pruned without a noticeable impact on performance.

(a) Evolution of accuracy on the validation set of **SST-2** when heads are pruned from BERT according to $I_h$.

(b) Evolution of Matthew's correlation on the validation set of **CoLA** when heads are pruned from BERT according to $I_h$.

(c) Evolution of F-1 score on the validation set of **MRPC** when heads are pruned from BERT according to $I_h$.

(d) Evolution of the BLEU score of our **IWSLT** model when heads are pruned according to $I_h$ (solid blue).

Figure 6: Evolution of score by percentage of heads pruned.