[Reviews · NeurIPS 2019]

Reviewer 1



Originality: The work is fairly original, and certainly garnered a fair amount of attention when released on arXiv. I don’t think most MT researchers would have thought to try to drop attention heads after training is complete. But this is obviously a very familiar idea to those working in network pruning . Quality: The experimental work here is top-notch, experiments are well-designed and described clearly, with statistical significance clearly indicated when appropriate. However, it is a little disappointing that one of the “soundbite” results from this paper, that “some layers can be reduced to a single head” was (a) an oracle result where the most important head was selected by looking at test set and (b) was done one layer at a time. My main takeaway from the paper, that MT/BERT Transformers can safely prune 20/40% of their heads, is much less surprising and exciting. The second technical contribution mentioned above is fairly minor and not particularly novel; the paper should be seen as being mostly experimental, with much of its novelty being derived from the fact that it is covering ground that is as of yet under-explored in NMT/NLP. Clarity: Aside from the above mentioned disconnect between the abstract/conclusion and the rest of the content, the paper is exceptionally clear. I suppose I also didn’t find Figure 5 to be as clear and helpful as the authors found it - it is hard to see linear relationships between results at a given timestep, and it is not clear why the authors choose to call out Epoch 10 as being particularly important (as opposed to Epoch 6, for example). Significance: This paper could potentially start a trend in trying to prune NMT and BERT systems. But I’m not sure any major players will be rushing to prune their attention heads based on these results as is. Specific questions: Equation (1) shows attention heads being combined in a sum. Aren’t they usually combined by concatenation? ====== Thanks to the authors for their response. I greatly appreciate how much space and attention was devoted to my review. The clarifications to Figure 5 and to Equation (1) are helpful. The additional experiments showing greater reductions from pruning certain BERT models, and the generalizability of the "one head" claim across dev and test are very comforting. I have adjusted my score accordingly. Having now read the other reviews, I sympathize with R2's concerns about how good of a fit this work is for a NeurIPS audience. Thinking about this from an ML (as opposed to NLP) standpoint, it would have been interesting to see how much more efficient pruning attention heads is than pruning network nodes without structural constraints. I imagine it is certainly easier to get speed-ups with this structured pruning.

Reviewer 2



This paper prunes the heads for transformer models and empirically asks whether they are needed. The answer seems to be that we don't need many heads for good accuracy. This is an interesting empirical result, but I think there are a few more experiments that should be run to convince the reader that the conclusions are general: (a) Repeating the analysis with transformer modeled trained with different number of heads. (b) Repeating the analysis on more datasets, e.g. transformer trained on a different dataset. I understand there was an IWSLT experiment in Sec 6 but it asks slightly different problems than Sec 2-5. Clarification questions: - Fig 1. y axis is number of heads, which is a bit confusing. Is it supposed to be frequency of models with a given BLEU/accuracy instead? Or is this plot really binnned by number of heads? - Fig 3. What is the green line? I don't understand why pruning based on the end metric (BLEU or accuracy) would do worse than the blue line (I_h)?

Reviewer 3



This paper offers a solid discussion of pruning attention heads in models using multi-headed attention mechanisms. The provided 'heuristic' strategies to do so seem effective, yet one could imagine additional variants worth evaluating. The analysis is solid, the findings somewhat surprising and practically highly relevant as they improve inference speed considerably.

[Author Response · NeurIPS 2019]

**General comments** We thank the reviewers for their insightful feedback. First we would like to apologize for, and correct an inaccuracy in our inference speed experiments in the submitted draft. A hardware default on our end caused abnormally slow inference for some experiments. We have rerun the speed comparison in a more robust setup (6 averaged runs across 2 different machines with GTX 1080 Ti GPUs). The updated results are as follows:

**Reviewer 1** We understand the reviewer's main concern to be the impact of our results given that the experiments in section 3.3 are performed with an oracle pruning approach, and that systematic pruning allows to prune only from 20%/40% of heads.

| Batch size | 1 | 4 | 16 | 64 |
|---|---|---|---|---|
| Original | $17.0 \pm 0.3$ | $67.3 \pm 1.3$ | $114.0 \pm 3.6$ | $124.7 \pm 2.9$ |
| Pruned (50%) | $17.3 \pm 0.6$ | $69.1 \pm 1.3$ | $134.0 \pm 3.6$ | $146.6 \pm 3.4$ |
| | (+1.9%) | (+2.7%) | (+17.5%) | (+17.5%) |

Table 1: Average inference speed of BERT on the MNLI-matched validation set in examples per second ($\pm$ standard deviation). The speedup relative to the original model is indicated in parentheses.

**Oracle pruning in section 3.3** As we understand it, the reviewer's issue with the sections is that the best "only one head" scores for each layer in tables 2/3 are both chosen and reported on the same dataset. While we realize this may be subjective, even in this "oracle" setting we were surprised by the fact that some layers only need a single head (and others to whom we have conveyed the results expressed similar opinions). Nevertheless, based on this comment we additionally performed experiments to choose the best "single layer" on a validation set (newstest2013/MNLI train subset) and report the scores on a test set (newstest2014/MNLI dev set). In particular for newstest2013/14 we find that for more than half of the layers, we can pick a head on the dev set such that keeping only this head results in a change of BLEU score that is not significant on the test set. We also notice similar patterns as Table 2 in the paper, e.g. this phenomenon is much more present in Dec-Dec attention, whereas Enc-Dec attention suffers much more from keeping only one head (-18.89 BLEU for the last layer). The detailed table will be included in the final version of the paper, and we will clarify and contextualize the "only one head is sufficient" claim in the abstract, introduction and conclusion.

**Impact** We see two issues here. First, the reviewer suggested the total percentage that can be pruned without decreasing performance is too low (20/40%) to be of general interest. Regarding this, since submission we have performed experiments on additional GLUE tasks (SST-2, MRPC, CoLA), and noticed that up to 60% of the heads could be pruned (in SST-2, see fig. 2). Second, the reviewer commented that this paper is unlikely to interest "major players". In counterpoint, we would like to note that the proposed method has already been independently re-implemented (by a third-party) in a popular open source library with over 10,000 stars on github (for anonymity purposes, we will not specifically cite it here).

**Clarity of figure 5** We propose to clarify the phenomenon by supplementing Figure 5. with the plot on the right, showcasing the relationship between percentage pruned and percentage BLEU lost for the first two epochs and at the end of training (epochs 35 and 40). We will add this to the revised paper.

**Sum vs. concatenation notation for MHA in eq. (1)** We apologize for any confusion caused here. This notation is equivalent to the concatenation formulation since concatenating then multiplying by a $d \times d$ matrix is equivalent to multiplying by $h$ $d \times (d/h)$ matrices. We used this formulation to ease exposition of the masking variables, but will clarify this in the revision. Thank you for noting that this was unclear.

**Reviewer 2** As we understand, the reviewer's main issue with the paper is the number of models/datasets tested, and the overall significance of the results.

**Number of models/datasets** We had already performed additional experiments that, for reasons of space, have been left out from the submitted version. Specifically, we have performed experiments on 3 additional GLUE tasks (SST-2, MRPC and CoLA) and on IWSLT de-en, obtaining similar results as MNLI and WMT (see eg. SST-2 in Fig. 2), which we intend to include in the final version.

**Significance** We respectfully disagree with the reviewer that our contribution lacks generality. As pointed out in the paper, multi-head attention based models are ubiquitous in state-of-the-art NLP (MT, BERT, XLNet...) and other domains (e.g. "A Time-restricted Self-attention Layer for ASR" for speech). The two types of models we have experimented are among the most widely used versions (BERT and Transformer-based MT).

**Reviewer 3** The main concern seems to be the pruning method. We agree that previously proposed pruning methods may yield different advantages, but stress that the main point of the paper is elucidating the fact that particular attention heads can be completely removed from the model without serious negative effects. Also in contrast to many previous approaches, removing full attention heads at inference time is both efficient and simple, which we view as a strong point of the proposed approach.

[Meta-Review · NeurIPS 2019]

This work investigates whether multi-headed attention models (e.g., BERT) actually need to use multiple attention heads. The perhaps surprising finding is that in some cases a single head suffices. Reviewers agreed that the question here is interesting and the empirical work sound. This work may motivate follow-up efforts that investigate similar pruning exercises.